# Complex Emulsions as an Innovative Pharmaceutical Dosage form in Addressing the Issues of Multi-Drug Therapy and Polypharmacy Challenges

**DOI:** 10.3390/pharmaceutics16060707

**Published:** 2024-05-24

**Authors:** Naresh Yandrapalli

**Affiliations:** Max Planck Institute of Molecular Cell Biology and Genetics, Pfotenhauerstrasse 108, 01307 Dresden, Germany; yandrapalli@mpi-cbg.de

**Keywords:** complex emulsions, microfluidics, polypharmacy, drug delivery, multi-drug therapy

## Abstract

This review explores the intersection of microfluidic technology and complex emulsion development as a promising solution to the challenges of formulations in multi-drug therapy (MDT) and polypharmacy. The convergence of microfluidic technology and complex emulsion fabrication could herald a transformative era in multi-drug delivery systems, directly confronting the prevalent challenges of polypharmacy. Microfluidics, with its unparalleled precision in droplet formation, empowers the encapsulation of multiple drugs within singular emulsion particles. The ability to engineer emulsions with tailored properties—such as size, composition, and release kinetics—enables the creation of highly efficient drug delivery vehicles. Thus, this innovative approach not only simplifies medication regimens by significantly reducing the number of necessary doses but also minimizes the pill burden and associated treatment termination—issues associated with polypharmacy. It is important to bring forth the opportunities and challenges of this synergy between microfluidic-driven complex emulsions and multi-drug therapy poses. Together, they not only offer a sophisticated method for addressing the intricacies of delivering multiple drugs but also align with broader healthcare objectives of enhancing treatment outcomes, patient safety, and quality of life, underscoring the importance of dosage form innovations in tackling the multifaceted challenges of modern pharmacotherapy.

## 1. Introduction

The phenomenon of polypharmacy characterized by using multiple medications by a patient, often concurrently to manage several conditions, has emerged as a significant challenge in contemporary healthcare [1,2]. This practice is especially prevalent among the elderly population, where multimorbidity necessitates the administration of multiple drugs to manage distinct conditions effectively [3]. While necessarily for comprehensive care, both MDT and polypharmacy introduces complexities, including increased risks of adverse drug incompatibilities, potential drug–drug interactions, and challenges in medication adherence, all of which can compromise patient safety and treatment outcomes. The consequences of polypharmacy also lead to the patient’s non-adherence to treatment due to pill-burden, i.e., the number of dosage forms consumed per day [4].

In response to these challenges, advancements in pharmaceutical technologies, particularly through the development of microfluidic systems and complex emulsions, present a novel approach to drug delivery [5,6]. Microfluidic technology leverages the manipulation of fluids at a microscale range to produce complex emulsions with unparalleled precision in droplet size, composition, and encapsulation efficiency [7,8]. These systems enable the simultaneous encapsulation of multiple drugs within a single delivery vehicle, offering a strategic solution to the pill burden associated with polypharmacy [9,10]. By allowing for the controlled release of each drug at specific sites and times, microfluidic-generated complex emulsions can potentially enhance therapeutic efficacy, reduce side effects, and improve patient compliance.

Microfluidics-based devices are already playing a crucial role in developing advanced drug delivery [11]. The integration of these technologies into drug delivery systems aligns with the broader goals of personalized medicine, offering tailored treatments that consider individual patient needs, conditions, and responses to therapy [12]. As such, microfluidic complex emulsions not only represent a technical advancement in the field of pharmaceutical sciences but also provide a patient-centric approach to addressing the growing concerns associated with polypharmacy. This review seeks to explore the intersection of microfluidic technologies and complex emulsion development, emphasizing their potential to revolutionize multi-drug delivery and mitigate the drawbacks of current polypharmacy practices. Thus, the review briefly describes droplet microfluidics and their advantages over traditional methods of emulsion making, followed by an in-depth analysis of double and complex emulsions in multi-drug delivery. At the end, the review will present existing challenges and possibilities to establish complex emulsion-based formulations as a dosage form in real-world use cases. Acknowledging the plethora of reviews available in microfluidics-based emulsion technology—readers are directed to existing reviews from the literature. For example, multiphase fluid flow in microfluidics, as discussed by Zhao et al. (2013) and Baroud et al. (2004), deals with the fundamentals of fluid flow in microchannels [10,13]. Regarding microfluidic production of multiple emulsions, Vladisavljević et al. (2016 and 2017) discusses controlled production mechanisms of multiple emulsions [14,15]. Moreover, microfluidic emulsion techniques for emulsion and microcapsule synthesis methods are discussed by Wang et al. (2023) [10,15,16,17]. Thus, this review is primarily dedicated to the idea of realizing complex emulsions-based, multi-drug delivery solutions using microfluidics, for brevity, while presenting a novel perspective in the fascinating field of emulsion technology in pharmaceutics. The below schematic represents the theme of this review where pill-burden-stricken, multimorbid patients can be alleviated via a microfluidic-based complex emulsion solution—combining four or more pills/active ingredients into one compartmentalized emulsion system (Figure 1). Henceforth, the existing state of the art, further possibilities and challenges, and gaps in the literature will be presented. The review is structured in a manner to present droplet microfluidics and the emulsions derived for applications in multi-drug delivery use cases, i.e., co- and multiple encapsulations of actives within double and other complex emulsions. Thereafter, the major challenges will be described in detail including stability, leakage, formulation, and scalability requirements. In the end, conclusive thoughts and next steps will be presented.

## 2. The Rise of Droplet Microfluidic Technologies in Drug Delivery

The advent of droplet microfluidic technologies has marked a significant leap forward in the realm of drug delivery—enabling the precise manipulation of fluids at the microscale to create emulsions tailored for drug encapsulation and release [7,10,11]. At the heart of this advancement is the ability to control droplet formation, chemical composition, and stability with unprecedented precision, offering new avenues for therapeutic interventions.

### 2.1. Principles and Advantages of Droplet Microfluidic Systems

Microfluidic systems operate by manipulating small volumes of fluids within channels of micrometre dimensions (see Figure 2). These systems leverage various physical mechanisms, such as laminar flow and flow-focusing, amongst others, to produce droplets, particles, and emulsions with high uniformity and reproducibility. In simple terms, the continuous phase (fluid that can wet the microchannels) is allowed to shear the dispersed fluid (cannot wet the microchannels), leading to the production of spherical droplets due to surface tension.

Figure 2a presents the schematic representations of different microfluidic geometries that are in use to produce droplets. All the different geometries presented in Figure 2—T-junction, co-flow, and flow-focusing—can be employed to produce single emulsions [18,19,20,21] as well as double emulsions [22,23] and other complex emulsions [24,25] when used in series. For example, a single T-junction-containing microfluidic device can produce water-in-oil (W/O) or oil-in-water (O/W) single emulsions (Figure 2a). However, when two T-junctions are implemented in series, it is possible to achieve double emulsions like water-in-oil-in-water (W/O/W) (Figure 2b). Figure 2b also presents the schematics of various higher-order emulsions that are of importance in the community for their diverse applications, beyond what is offered by single emulsions.

From double emulsions to high-order emulsions, alternating layers of immiscible fluids form distinct compartments, which can be dispersed with active ingredients. These emulsions can be categorized under multiple or complex emulsions. The key advantages of using microfluidic technologies in producing complex emulsions over other methods such as membrane emulsification (detailed review on membrane emulsification-based production of complex emulsions can be found here [26,27,28]) are presented hereafter.

(a)Precision and Control:

Microfluidic technologies have ushered in an era of unprecedented precision and control in the realm of droplet generation over traditional homogenization techniques [8]. Fluid flow in microchannels enables the manipulation of fluids on a microscopic scale, allowing researchers to produce droplets, particles, and emulsions with highly precise dimensions and compositions [20,22]. This level of control is critical for ensuring consistent drug encapsulation and release profiles to match the pharmacokinetic requirements at each administration [6]. The ability to fine-tune the characteristics of the delivery vehicle at such a granular level significantly enhances the efficacy and safety of the dosage forms. Our own research suggests that microfluidics can be used to produce double emulsions at high precision in both size and encapsulation efficiency with RSD values sub-5% [22,29]. Figure 3a,b presents the liposomes generated after double emulsion templating, showing narrow distribution in both size and encapsulate fluorescence intensity. This is unprecedented and not possible through bulk methodologies such as the inverted emulsion method [30,31] or membrane emulsification techniques [26,27]. These results support reproducible precision using microfluidic methodology.

(b)Versatility:

The versatility of microfluidic systems stems from their ability to handle a wide range of materials and formulations, from aqueous solutions to volatile organic solvents (limited to glass-based microfluidics), viscous polymers, nanoparticles [20,33,34], and sensitive biological entities like genetic material [22,35], enzymes [36,37], and even cells [22,23,35]. From Figure 3c it can be understood that a droplet microfluidic platform can be used to encapsulate a wide range of materials such as plasmid DNA, small unilamellar vesicles, cells, and nanoparticles [20,22]. Moreover, microfluidics can be adapted to incorporate various active pharmaceutical ingredients, enabling the possibility to co-delivery multiple drugs with distinct physicochemical properties within a single platform—for example, to be able to include both hydrophilic and hydrophobic drugs within the same system. By allowing poly (*N*-isopropylacrylamide) (pNIPAm) microgels to encapsulate, retain, and systematically release therapeutic agents of diverse solubilities with high efficiency, Jagadeesan et al. has shown that multi-drug therapy can be made possible with emulsions developed using microfluidics [38]. This adaptability opens new possibilities for combinatorial therapies, where the synergistic effects of different drugs can be harnessed to enhance treatment outcomes. This flexibility to encapsulate a wide range of materials within emulsions makes microfluidics a powerful platform for developing novel drug delivery vehicles.

### 2.2. Co-Encapsulation in Double Emulsions

The application of droplet microfluidic technologies in creating complex emulsions represents a transformative approach to drug delivery systems, offering unique capabilities that traditional methods cannot match [10,11,39]. Complex emulsions, such as double emulsions, water-in-oil-in-water (W/O/W), or oil-in-water-in-oil (O/W/O), can be precisely engineered to encapsulate multiple therapeutic agents, including those with differing solubility profiles, within distinct phases of the emulsion. A hydrophobic drug can be dispersed within the oil phase of the emulsion while the hydrophilic drug can be dispersed in the aqueous phase. A singular double emulsion can accommodate two hydrophilic drugs and one hydrophobic drug. Of course, through polymerization within the inner aqueous phase, middle oil phase, or both, one could control the release kinetics of these encapsulated drugs. In an interesting study, Francisca et al. (2015) presented a dual-delivery, complex, emulsion-based dosage form involving proteins, enzymes, and nanoparticles [24]. Poly (lacto-co-glycolic acid) (PLGA)-based nanoparticles loaded with glucagon-like peptide-1 are further coated with chitosan and cell-penetrating peptides, which are then dispersed in aqueous solution and sheared using dipeptidyl peptidase-4 (DPP4) and hydroxypropyl methylcellulose acetyl succinate containing ethyl acetate oil phase to form W/O droplets (see Figure 3d). This suspension is used as an inner solution to form complex W/O/W emulsions. Thus, the authors were able to encapsulate drug-loaded nanoparticles within double emulsions and use them for oral drug delivery. Recently, in a similar study, Jiang et al. (2021) has shown burst and sustained release of curcumin and dexamethasone using a double-emulsion-based formulation to treat inflammatory bowel disease. As shown in Figure 3e, two types of double emulsion were made using either curcumin- or dexamethasone-loaded PLGA nanoparticles in the middle oil phase and curcumin or dexamethasone in the inner aqueous phase [32]. The burst release property is achieved through using a pH sensitive polymer—hydroxypropyl methylcellulose acetate succinate, which undergoes quick disintegration upon exposure to low pH. The sustainable release kinetics are due to the drug-loaded nanoparticles that provide prolonged drug release. Their work presented an interesting methodology to obtain burst and sustained release therapeutics, which might not be possible using single emulsions. In addition to pharmaceutical applications, W/O/W emulsions loaded with xanthoxylin, and vitamin C are presented as an effective antiaging formulation [40]. The co-encapsulation within W/O/W contains vitamin C in the aqueous phase and the hydrophobic active, xanthoxylin, in the middle oil phase. Later, the authors presented an interesting coacervation strategy to convert the produced W/O/W emulsions into microcapsules. This suggests that these compartmentalized W/O/W emulsions can not only be used for pharmaceutical purposes but also in cosmetic applications. Similarly, a significant amount of work has been conducted in producing W/O/W emulsions for co-encapsulation of various active ingredients [41,42,43]. Microfluidic technology is not the only methodology employed to produce W/O/W double emulsions for this purpose. Membrane emulsification and agitation methods to encapsulate actives within these emulsions are other alternative methods. As listed in Table 1, it evident that dual drug encapsulation is also possible through these methods; in fact, these methodologies can produce larger volumes of double emulsions in short span when compared to microfluidics [42,43,44,45] (refer to scalability under Section 4). However, both methods have limited dosage control due to lack of precision encapsulation unlike microfluidics. Although, most of the works using these alternative methods reported narrow size distribution of the overall emulsion, categorical study of the dosage delivery per volume of double emulsion is more important in real-world use case scenario. Even if the discrepancy is low, dosage accuracy is highly valuable when potent and expensive active ingredients are used in the formulation. In addition to this, the dual drug co-encapsulation strategies can be replicated using W/O emulsions itself and the studies have not truly utilized the three distinct compartments offered by the W/O/W double emulsions. In the Table 1, the outer fluid (OF) is devoid of active ingredients. For example, inclusion of antioxidants in the OF can be protect the degradation of components within inner fluid (IF) and middle fluid (MF) compartments.

Furthermore, it is possible to increase or decrease the size of the inner droplet or the overall emulsion size, thus providing control over the concentration of the aqueous or lipo-soluble drug dosage. This control results in desired uniform droplet sizes and compositions, essential for ensuring consistent therapeutic outcomes. In our own research, we have proven this through varying the middle oil phase of the double emulsion. By altering the flowrates, we managed to produce double emulsions with thicker and thinner middle oil phases loaded with nanoparticles [20]. In a different study, we have also shown that, through careful manipulation of flow rates, it is possible to reduce the middle oil phase thickness [22]. Moreover, the ability to tailor the droplet interface properties through the selection of surfactants or polymers enables the stabilization of the emulsions, preventing coalescence and ensuring long-term stability of the encapsulated active ingredient [46] (refer to stability in Section 4).

The versatility of microfluidic platforms facilitates not only the encapsulation of drugs but also the incorporation of targeting moieties, such as antibodies or peptides, onto the surface of the droplets [47,48]. Through surfactant tagging the antibodies or proteins one could target the emulsion to specific tissue within the gastro-intestinal tract. This targeting capability enhances the delivery of drugs while improving the efficacy of the therapy and minimizing side effects [12].

However, to our surprise, there is almost no literature available about incorporating more than two active ingredients within double emulsion microfluidics. Furthermore, we are unable to find any data with active ingredients in the three compartments a double emulsion offers. This suggests that the field has not progressed beyond converting double emulsions into microcapsules, with no interest in taking advantage of the distinct compartments of a double emulsion. This brings us to the next step—microfluidic emulsions with more than three distinct compartments, nested and other complex emulsions.

**Table 1 pharmaceutics-16-00707-t001:** Co-encapsulation of active ingredients within W/O/W double emulsions.

Method	Inner Fluid (IF)	Middle Fluid (MF)	Outer Fluid (OF)	Advantages	Reference
Membrane emulsification	A total of 10 mg Epirubicin, 5.8% (*w*/*v*) glucose solution	A total of 500 mg tetra-glycerine-condensated ricinoleate in iodized poppy seed oil	A total of 5 mL of normal saline and 50 mg of poloxamer-188.	Uniform size with large volume production capabilities	[49]
	^10^BSH in 5% glucose solution	Iodized poppy seed oil with polyoxyl-40 castor oil (HCO40)	Saline solution with surfactant	No side effects with facile preparation methodology	[50]
	Vitamin B12 in 0.1 M NaCl	Trans-Resveratrol in 20% ethanol	Miglyol 812 with 5% PGPR and 0.5% CMC in 0.1 M NaCl	High stability and feasible preparation	[44]
Agitation	Vitamin C with NaCl and gelatine in deionized water	Xanthoxylin (GX-50) in propylene glycol with 5% PGPR containing Olive oil	GA-NaCMC (6:1) in deionized water	Complex coacervation based synthesis of microcapsules	[40]
	Amounts of 0.584 g NaCl, 0.04 g sodium azide, 0.441 g sodium caseinate, and 44.1 mg Hydroxytyrosol dissolved in 100 mL distilled water	Amounts of 94% perilla oil with 6% PGPR	Amounts of 0.584 g NaCl, 0.04 g sodium azide, and 0.5 g sodium caseinate in 100 mL distilled water	Gelation-based stabilization of emulsions	[51]
	A total of 20%, *w*/*w* Insulin in 0.1 mol/L HCl pH 7	1 mg/mL Quercetin with 5% PGPR in soybean oil	A total of 2% (*w*/*v*) Tween 80 or lectin or pectin	Increased stability and sustained release	[43]
	Amounts of 80 mg Epigallocatechin-3-Gallate in 2% (*w*/*v*) saccharose and different levels of gelatine (0%, 1%, 3%, 5%, and 10% *w*/*v*) dispersed in distilled water adjusted to pH 5.5.	A total of 40 mg/mL Quercetin ethanol solution with 5% PGPR in corn oil	A total of 1.5% (*w*/*v*) of gliadin nanoparticles in distilled water adjusted to pH 5.5	Controlled release and increased bioaccessibility	[42]
	A total of 500 μg/mL of arbutin, 3% of gelatine (*w*/*v*) and 3% of NaCl (*w*/*v*)	A total of 500 μg/mL of coumaric acid and 8% of PGPR (*v*/*v*) in olive oil	Whey protein concentrate (WPC) alone, WPC–gum Arabic (GA) or WPC–high methoxyl pectin (PEC) complex	Protein–polysaccharide complex stabilization and Sustained release properties	[45]
	A total of 0.5 mg mL^−1^ Vitamin B2	Amount of 20% (*w*/*w*) Vitamin E with 8% PGPR in soybean oil	WPC isolate or WPC–pectin or WPC–k-carrageenan	Controlled release with pH resistance	[52]
	Amounts of 10.0 wt.% Collagen peptide, 0.6 wt.% Flaxseed gum (FG)in 100 mM NaCl solution.	Amounts of 0.1 wt.% Astaxanthin, rice barn wax (RBX) (0–7.0 wt.%), and 2.0 wt.% PGPR in soybean oil	FG/WPI complexes (1.0 wt.%) in 100 mM NaCl	Increased bioavailability and chemical stability	[53]
Microfluidics	Glucagon-like peptide-1 20 mg of PLGA + CS-CPP or 200 μg of PSi + CS-CPP	A total of 0.5 mg dipeptidyl peptidase 4 of DPP4 inhibitor was added into 1 mL of 4% of HPMC-AS dissolved in ethyl acetate	A total of 2% Pluronic F127 in sucrose solution (100 mOsm L^−1^)	Controlled release with improved bioavailability and retention	[24]
	A total of 2 mg/mL Doxorubicin Hydrochloride	A total of 67 μg/mL Paclitaxel in molten lipid (Witepsol H15, Sasol)	A total of 10 wt.% PVA	Solvent-free encapsulation with biodegradable carrier	[54]
	Nile Red in Hexadecane	DAPI with monomer *N*-isopropylacrylamide (NIPAm, 10 wt.%), the crosslinker *N*,*N*′-methylenebisacrylamide (BIS, 0.2 wt.%), the surfactant Brij 35 (5 wt.%), and the photoinitiator 2,2′-azobis (2-methylpropionamidine) dihydrochloride (0.08 wt.%)	A total of 4 wt.% Span 83 in soybean oil	Stimulus-responsive delivery of multiple drugs	[38]
	Catechin (1 mg/mL) was dissolved in water pH 3	Nile Red PDMS_60_-*b*-PDMAEMA_50_ copolymer (5 mg/mL) was dissolved in 3 mL of oil (miglyol^®^812, or isopropyl myristate)	Sucrose (1 mg/mL) were dissolved in water	Low pH drug encapsulation	[55]
	A total of 3 mg Indocyanine Green (ICG) and 2% (*w*/*v*) synthesized DOX–ADA solution	10% (*w*/*v*) PLGA solution in dichloromethane	A total of 10% (*w*/*v*) PVA in ultrapure water	Chemo- and photothermal response with sustained drug release	[56]
	Silver nanoparticles 0.5% sodium alginate and 10% (*w*/*v*) PVA	Oil Red O with DCM with 20% (*w*/*v*) PLGA (50:50)	A total of 10% (*w*/*v*) PVA and 4% calcium chloride (CaCl_2_)	Core-shell-structured microparticles	[57]
	A mixture of soybean oil and benzyl benzoate (1:1, *v*/*v*) containing free curcumin/ (3 mg/mL), Rhodamine B-loaded PLGA nanoparticles (3 mg/mL), terephthalaldehyde (2.4 wt.%), and PGPR (8.0%, *w*/*v*)	Chitosan (2.0%, *w*/*v*), F127 (1.5%, *w*/*v*), and HEC (2.0%, *w*/*v*)	Soybean oil containing PGPR (8.0%, *w*/*v*)	Sequential drug release with burst release and sustained release profiles	[58]
	Doxorubicin (1 mg mL^−1^) and 10% sucrose	Oregon-Green-dye-labelled paclitaxel (green, 40 µg mL^−1^) in the soybean oil layer	A total of 10% PEGDA, 2% PVA, and photoinitiator/2% Span80 in mineral oil	Multi-stimuli such as chemical dissolution, mechanical stress, and osmotic pressure	[59]
	Curcumin-loaded PLGA nanoparticles (20 mg mL^−1^) or dexamethasone-loaded PLGA nanoparticles (20 mg mL^−1^) with 2 wt.% PEG aqueous solution	Ethyl acetate and dichloromethane (4 : 1, *v*/*v*) containing curcumin (300 μg mL^−1^), dexamethasone (300 μg mL^−1^), and enteric HPMCAS-HF (100 mg mL^−1^)	A total of 2 wt.% PVA aqueous solution	Sequential burst–sustained drug release	[32]
	FITC-Dextran pH 7.4	PLGA dissolved in chloroform or dichloromethane with the addition of Nile Red (100 µg mL^−1^)	A total of 2 wt.% PVA aqueous solution	Mechanical-stress-induced dual drug release	[60]
	A total of 15% (*w*/*v*) photo-crosslinkable GelMa, including the drug DOX hydrochloride (500 μg mL^−1^)	A total of 10% (*w*/*v*) PLGA and Camptothecine (500 μg mL^−1^) dissolved in DCM	A total of 2 wt.% PVA aqueous solution	Sustained drug release with improved cell viability	[61]

## 3. Compartmentalized Complex Emulsions for Multi-Drug Delivery

The implementation and application of complex emulsions through microfluidic technologies represents a cutting-edge approach in the field of multi-drug delivery systems. These emulsions, characterized by their ability to encapsulate multiple therapeutic agents within distinct compartments, offer a sophisticated solution to the challenges posed by MDT and polypharmacy. Research in the direction of double emulsions has clearly presented the idea that co-encapsulation or simultaneous drug delivery can be made possible. This section delves into more intricate complex emulsions, their potential significance in multi-drug delivery, and the benefits they bring to patient care.

### 3.1. Concept and Significance

The concept of complex emulsions as a dosage form transcends traditional drug delivery paradigms by offering a platform capable of encapsulating multiple drugs (more than three) within a single delivery vehicle. This innovation is pivotal for the pharmaceutical industry, addressing the growing need for more efficient and effective treatment strategies, particularly for diseases that require the administration of multiple therapeutic agents [3,62]. Like double emulsions, multiple emulsions such as nested emulsions, layered emulsions and high order emulsions, through their inherent design, can house different drugs within separate compartments or phases, thereby preserving the stability and integrity of each drug while enabling their simultaneous or sequential release [52,58]. For example, complex emulsions can be a water-in-oil-in-water system with more than one inner aqueous droplet (see Figure 2b and Figure 4a,b). In this case, the entire system can have a minimum of three distinct hydrophilic compartments (two inner aqueous droplets and one outer aqueous environment) and one lipophilic compartment, allowing for at least four different active ingredients to be encapsulated. This unique architecture can also enable concurrent or sequential release of different drugs from a single delivery vehicle, enhancing the efficiency of treatment regimens by providing synergistic or complementary therapeutic effects. Unfortunately, as is evident from Table 2, there has not been any significant progress in this direction. The data presented in Table 2 suggests that, in most cases, the encapsulated materials are only dyes; interestingly, in a few cases, the encapsulated materials are polymersomes [63], nanoparticles [25], and even cells [64]. Despite the lack of encapsulation of medicinal components, substantial work has been conducted using representative compounds such as fluorescent dyes and polymers.

Thus, it is worthwhile to examine these data. Figure 4b,c presents microfluidics-based designing of nested emulsions. Unlike the data presented in Figure 4a where there is no-difference in inner droplet content (all of them have identical chemical composition), Figure 4b,c has inner compartments with distinct fluorescent components. Most interestingly, the inner compartments in Figure 4b are dye-loaded aqueous solutions [65,66]. In Figure 4c, they are dye-loaded polymersomes within their aqueous solutions [63]. This is fascinating because using this construct the authors have shown sequential release of the model drugs [63,67]. Similarly, layered emulsions (capsule within a capsule) are presented as an interesting strategy to provide such sequential delivery of drugs [67].

Unlike double or nested emulsions, layered emulsions are unique as they are multiple layers of emulsions within emulsions. As presented in Figure 2b and Figure 4d,e, their production requires intricate microfluidic setup with multiple hydrophilic and hydrophobic regions ensuring proper wetting of the continuous phases and eventually leading to precise layer-by-layer geometries. In fact, Figure 4d,e presents different methodologies of producing layered emulsions. In the first case, Figure 4d, each layer or compartment is assembled in a stepwise fashion—one after the other, as shown in section (a) of Figure 4d. In section (b) of the figure, formed W/O droplets are pumped further to form W/O/W emulsions and later W/O/W/O emulsions; finally, a W/OW/O/W triple emulsion is produced—seen more clearly in section (c) of Figure 4d [68]. Similar strategies are also employed by multiple researchers to yield such intricate compartmentalized structures [17,69,70]. Interestingly it is also possible to produce such intricate designer emulsions via sequential loading of preformed double emulsions [67]/ liposomes [71,72]/ polymersomes [63,67] into a double emulsion device to achieve layered emulsion. This methodology allows the use of standard double emulsion device. Furthermore, it allows researchers to process or surface treat the primary double emulsion before encapsulating it within the secondary orouter double emulsion—thus making it feasible for interesting delivery strategies. In Figure 4e, the authors have prefabricated poly (ethylene glycol) (Mw 5000)-b-poly (lactic acid) (Mw 10,000) bilayer polymersomes using a double emulsion device and used them as an inner solution to create layered emulsions. In fact, the authors have produced triple polymersome microcarriers that can be triggered to release cargo in a sequential fashion, as seen in Figure 4f. This methodology of sequential release can play a significant role in polypharmacy, where drug–drug interactions are an issue (refer to drug interactions under Section 4). One could load such polymersomes with actives that need to be released in the stomach region within the outer polymersome, followed by another active to be released in the intestine region, and, finally, an active for the colon region within the inner-most polymersome. This could also be a strategy to avoid the degradation of actives in the presence of enzymes within a specific region of the gastro-intestinal tract.

**Table 2 pharmaceutics-16-00707-t002:** Compartmentalized emulsions for cargo release applications are produced using microfluidics.

Type of Emulsions	Formulation	Compartment One	Compartment Two	Reference
Nested emulsions	OF—47.5 wt.% glycerol and 5 wt.% PVA MF—molten fatty acid glycerides IF—50 wt.% glycerol	Wright stain	Rhodamine B	[73]
	OF—2 wt.% F108 and 10 wt.% PVA (2:1, *v*/*v*) MF—ETPTA with HMPP solution (1 v%) IF—2 wt.% F-108	Cell 1	Cell 2	[64,74]
	OF—PVA solution MF—chloroform and hexanes (36:64 *v*/*v*) with 10 mg mL^−1^ PEG(5000)-*b*-PLA(5000) IF—PEG solution	FITX-Dextran	PEG	[75]
	OF—10 wt.% PVA, (*M*_w_ 13,000–23,000) MF—chloroform and hexane (38:62, *v*/*v* %) with 5 mg/mL PEG-b-PLA IF—10 wt.% PEG (*M*_w_ 6000)	Polymersomes	Polymersomes	[63]
	OF—5% (*w*/*v*) PGPR90, soybean oil MF—1% (*w*/*v*) Pluronic F127, 5% (*w*/*v*) glycerol, 1.3% (*w*/*v*) NIPAM monomer, 0.77% (*w*/*v*) MBA crosslinker, and 0.6% (*w*/*v*) APS initiator. IF—5% (*w*/*v*) PGPR90, soybean oil	Sudan III	Unlabelled	[76]
	OF—0.5 wt.% F108 MF—Styrene—Octanol (95-5, *v*/*v*%) IF—0.5 wt.% F108	Gold nanoparticles	Gold nanoparticles	[25]
	OF—2% (*w*/*v*) PVA MF—10% (*w*/*v*) PLGA dissolved in DCM IF—15% (*w*/*v*) photo-crosslinkable GelMa	Doxorubicin	Camptothecine	[61]
Layered emulsions	OF 1—2 wt.% PEG, 8 wt.% PVA and 0.5 wt.% Pluronic^®^ F-68/OF 2—10 wt.% PVA with 1.0 wt.% F-68 MF—chloroform and hexane (36:64, *v*/*v*) containing 5 mg mL^−1^ egg PC IF—8 wt.% PEG and 2 wt.% PVA	Cell-free expression system	Cell-free expression system	[71]
	OF1 and OF2—A 10 w/w% PVA (Mw 13,000–23,000 g mol^−1^) MF—ETPTA with 0.2 w/w% HMPP and chloroform solutions containing either for MF 1—12 w/w% PLA (Mw 15,000 g mol^−1^) or for MF 2—17 w/w% PCL (Mn 45,000 g mol^−1^) IF1 and IF2—A 10 w/w% PVA (Mw 13,000–23,000 g mol^−1^)	8-hydroxyl-1,3,6-pyrenetrisulfonic acid, trisodium salt	Sulforhodamine B	[63,67]
	O_1_/W_2_/O_3_/W_4_/O_5_ quadruple emulsions W_2_ and W_4_ are deionized water with surfactant 0.5% (*w*/*v*) Pluronic F-127 and 10% (*w*/*v*) glycerine O_1_ and O_3_ are the mixture of soybean oil (SO) and benzyl benzoate (BB) with *V*_SO_:*V*_BB_ = 46:54, containing 2% (*w*/*v*) and 4% (*w*/*v*) PGPR, respectively; O_5_ and collection solution are SO containing 5% (*w*/*v*) PGPR	LR3000	Sudan Black	[68]
	O_1_/W_1_/O_2_/W_2_ triple emulsion O_1_—α-pinene in innermost oil phase W_1_—an aqueous solution of 2% PVA for the ultrathin water layer, O_2_—ETPTA as the second oil phase, and W_2_—aqueous solution of 10% PVA as the continuous phase	Fluorescein	Nile Red and α-pinene	[77]

**Figure 4 pharmaceutics-16-00707-f004:**
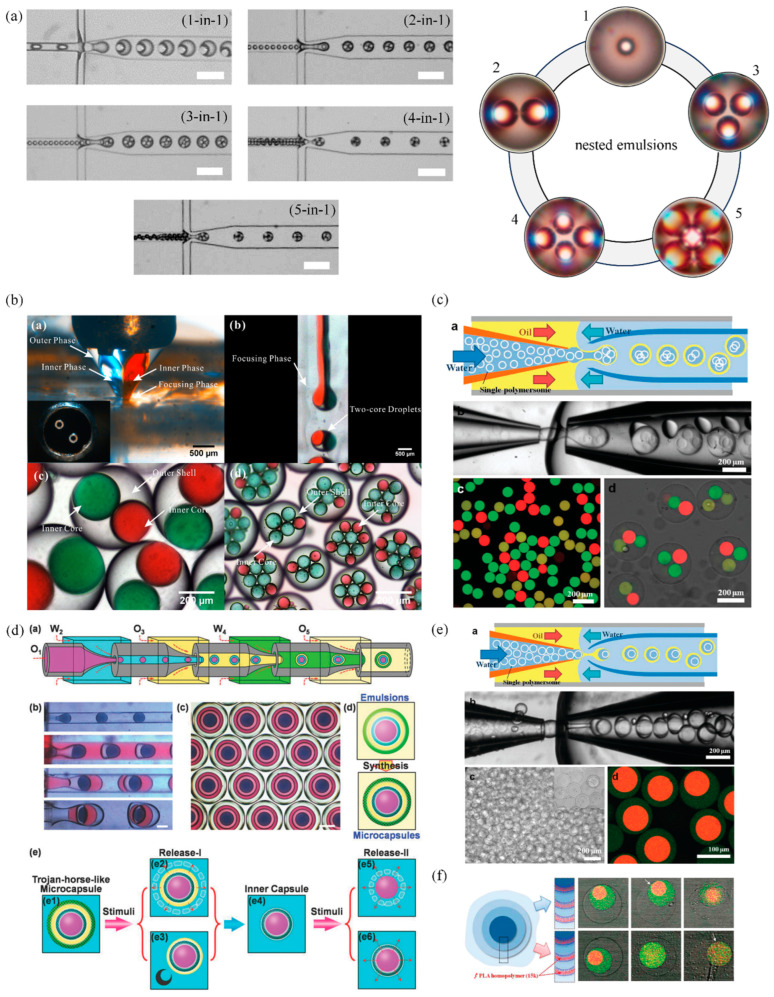
Nested emulsions. (**a**) PDMS-based production methodology for nested emulsions with increasing number of inner compartments, from one to five (all compartments are produced using same inner fluid). Inset showing the micrographs of the nested emulsions (Reprinted/adapted with permission from Ref. [78]. 2023, Naresh Yandrapalli). (**b**) Nested emulsions produced from two different inner solutions: red, Rhodamine B; green, Wright stain (Reprinted/adapted with permission from Ref. [65]. 2018, Elsevier). (**c**) Nested emulsions with prefabricated polymersomes with three different inner compartments (Reprinted/adapted with permission from Ref. [63]. 2011, American Chemical Society). (**d**) Layered emulsions with multiple hydrophobic and hydrophilic compartments produced using sequential layering of W/O to W/OW to W/O/W/O to W/O/W/O/W and their eventual conversion into double layered microcapsules with interesting cargo release properties (Reprinted/adapted with permission from Ref. [68]. 2018, Wiley and Sons). (**e**) Layered emulsions produced through encapsulation of preformed emulsions or polymersomes encapsulating different cargoes (shown in red and green colour) (Reprinted/adapted with permission from Ref. [63]. 2011, American Chemical Society). (**f**) Multi-layered emulsion showing triple membrane structure after polymerization, inset showing sequential disintegration of membranes leading to cargo release (Reprinted/adapted with permission from Ref. [63]. 2011, American Chemical Society).

The significance of complex emulsions in multi-drug delivery is profound, especially in the context of polypharmacy, where patients are required to manage multiple medications simultaneously. By combining several drugs into one formulation, complex emulsions reduce the pill burden on patients, potentially improving adherence to treatment regimens (see Figure 1). Moreover, these emulsions offer the opportunity to fine-tune the release profiles of encapsulated drugs, enhancing therapeutic outcomes by ensuring that each drug is released at the optimal time and location within the body, thereby maximizing its efficacy and minimizing side effects.

### 3.2. Mechanisms of Drug Encapsulation and Release

The mechanisms of drug encapsulation and release in complex emulsions are facilitated by the unique architecture of the emulsion. Microfluidics allows for the precise manipulation of multiple immiscible phases to form emulsions with highly controlled droplet sizes, compositions, and structures. This precision enables the encapsulation of drugs with vastly different physicochemical properties within the same particle—hydrophilic drugs in aqueous phases and hydrophobic drugs in oil phases, for instance.

Encapsulation is achieved through processes such as droplet formation in flow-focusing geometries or the breakup of multiphase streams in microchannel junctions, as shown schematically in Figure 1 [79,80]. These processes can be finely tuned to adjust the size of the droplets, the thickness of the membranes separating different compartments, and, thus, the concentration of drugs within each compartment.

The release of drugs from complex emulsions is governed by several mechanisms, including diffusion across the emulsion, degradation of biodegradable polymers used in the emulsion formulation [81], and external triggers such as pH changes [56], temperature fluctuations [54], or enzymatic activity. These mechanisms can be strategically designed to achieve desired release kinetics, such as sustained release [32,81,82] for chronic conditions or burst release [32,54,67] for acute interventions. Table 3 presents a list of research that has employed complex emulsions as drug delivery vehicles. From the table it is evident that a wide variety of stimuli are used as triggers to release the encapsulated actives.

Triggers include pH [32,56,68,83], temperature [54,66,76,80,83,84], osmotic pressure [25,59,81], hydrolysis [61,63], ultrasound [85], mechanical [60,81], photothermal [56], and photo switching mechanisms [86]. However, most research is confined to pH, temperature-, and osmotic-pressure-induced drug release. In Figure 5a, osmotic pressure is used to induce the release of actives, which is one of the common triggers for releasing cargo [10,12,81,87]. Interestingly, the delivery vehicles are made susceptible to both hypotonic and hypertonic osmotic pressure [81]. The researchers pointed out the buckling of the PLGA membrane under positive osmotic pressure difference. They pointed out that local failure of the membrane due to heavy folding under hypertonic conditions accelerates the release of encapsulant in comparison with the isotonic condition, although the rate is lower than the hypotonic condition (seen in panel (d) of Figure 5a). This work also includes an in vivo study, presenting sustained drug release in mice. These results showcase the potential of complex emulsions for in vitro and in vivo applications. One of the most curious triggers for drug release in complex emulsions is reported by Guo et al., 2024 (Figure 5b) [86]. The authors designed a photo-switchable block copolymer, PEG-PSPA that is solubilized along with PLGA polymer within the middle oil phase of the double emulsion. Under UV illumination, the spiropyran (SP) moiety transforms into merocyanine (MC), ionic form, enhancing the permeability. This reversible switching of SP to MC under UV–Vis light allows for on and off release of the encapsulated drug molecules (panel (d) and (e), Figure 5b). Thus, showing a fully regulatable burst release of drugs.

Beyond double emulsions, nested and layer-by-layer emulsions will have a rather complex release kinetics depending on the nature of inner cores/droplets and alternating hydrophilic and hydrophobic layers. One interesting study involving W/O/W/O/W presented an interesting mechanism of drug release [67]. Using glass capillaries, the authors produced a microcarrier within a microcarrier along with a model drug in each aqueous phase (Figure 5c). In the top two panels of Figure 5c, the membranes of both the capsules were made of poly (lactic acid) (PLA) that can undergo hydrolysis and resulted in a sequential release of model drugs, first from the outer capsule and then from the inner capsule. Panel (b) clearly depicts the burst release profile of the model drugs after PLA degradation. However, for the bottom panels (c) and (d) of Figure 5c, the outer capsule is made of polycaprolactone (PCL), which is partially resistant to hydrolysis, unlike the inner capsule, which is made of PLA. From the panel (d) in Figure 5c, it can be understood that the dye releases from the inner PLA carrier and mixes with the model drug present in the outer PCL carrier. Eventual hydrolysis of PCL resulted in slow and gradual drug release. Together, the authors have shown a layered emulsion system that can delivery drugs sequentially as well as simultaneously. Such strategies will greatly enhance the user applications of layered emulsions in polypharmacy where drug–drug interactions need to be avoided and site-specific drug delivery is needed.

Unlike single emulsions, drug release kinetics in complex emulsions is complex due to the multiple compartments made of opposing fluid phases that can be polymerized to form (semi)permeable membranes. An excellent study conducted by Dluska et al. (2017) presented experimental and mathematical modelling of drug release profiles within multiple emulsions (nested emulsions) using doxorubicin (DOX) as an anticancer drug [9]. They performed mathematical simulations to understand the diffusional release of DOX from multiple emulsions. The results have confirmed the possibility of modelling and predicting the release profiles in complex emulsions. A more detailed work on modelling drug release from double emulsions was also reported. Pontrelli et al. (2020) performed numerical simulations on double emulsion structures that were oblate and round. They found that such a structural change can affect the drug kinetics—the oblate shape exhibits faster drug delivery, while a round geometry promotes a more sustained release. At the micron scale, the gravitational force can result in sedimented inner droplets within the W/O/W emulsion—the inner droplet sits at the bottom of the middle oil phase resulting in the formation of a thin membrane that can be devoid of any oil [25,88]. Thus, the diffusional models proposed above so far are not entirely accurate considering this gravitational phenomenon. One could expect a much faster release of drugs along the thin membrane. Moreover, this thin membrane could be the main region of drug release depending on the integrity of the membrane, which, in turn, is based on the surfactants used to stabilise the emulsion. Nevertheless, increased interest in modelling drug diffusion in complex emulsion is essential to understand their passive release properties during extended shelf-life.

**Table 3 pharmaceutics-16-00707-t003:** Drug release profiles and triggering mechanisms implemented in complex, emulsion-based drug delivery vehicles.

Emulsion-Type	Release Profile	Trigger	Active	Reference
W/W/W	Sustained	Diffusion	Streptavidin and platelet-derived growth factor-BB	[82]
W/O/W	Burst	Temperature	Doxorubicin and Paclitaxel	[54]
W/O/W	Burst and sustained	pH	Curcumin and Dexamethasone	[32]
W/O/W	Sustained	Osmotic pressure and Mechanical	Indocyanine green and Nile Red	[81]
W/O/W	Sustained	pH (or) Photothermal	Doxorubicin & Indocyanine green	[56]
W/O/W	Prolonged	Hydrolysis (or) Mechanical	transforming growth factor-β3	[60]
W/O/W/O	Burst	Osmotic pressure	Doxorubicin and Paclitaxel	[59]
W/O/W/O/W	Burst	Hydrolysis (or) Osmotic pressure	8-hydroxyl-1,3,6-pyrenetrisulfonic acid, trisodium salt & sulforhodamine B	[67]
W/O/W/O/W	Burst	pH and Temperature	LR3000 and Sudan Black	[68]
W/O/W	Sustained	Hydrolysis	Doxorubicin and Camptothecine	[61]
W/O/W	Continuous (or) On and off	Photo responsive	Doxorubicin	[86]
W/O/W	Burst	Ultrasound	hexametaphosphate	[85]
O/W/O	Sustained	pH (or) Temperature	Vitamin B12	[83]
W/O/W	Continuous (or) On and off	Temperature	FITC-Dextran	[84]
W/O/W/O/W/O/W	Burst	Degradation	8-hydroxyl-1,3,6-pyrenetrisulfonic acid, trisodium salt	[63]
W/O/W	Burst	Temperature	RITC-Dextran	[80]
Nested W/O/W	Accumulated	Temperature	FITC-Dextran and RITC Dextran	[66]
Nested W/O/W	Burst	Temperature	Inner core release	[76]
Nested W/O/W	Burst	Osmotic pressure	Gold nanoparticles	[25]
O/W/O	Sustained	Glucose	FITC-Insulin (or) Rhodamine B	[89]

## 4. Challenges and Possibilities in Designing Complex, Emulsions-Based Formulations

With no real commercial formulations, it is worthwhile to discuss the major concerns that are stalling complex emulsions from seeing the light of the day, not only in pharmaceuticals but also in other commercial applications such as cosmetics and in the food industry.

Below, we discuss the possible hurdles and potential strategies to mitigate those hurdles.

(a)Leakage of Drugs:

Passive diffusion of drugs is the major culprit for drug leakage in emulsions. It is the principal transmembrane process for many small drugs. According to Fick’s law of diffusion, drug molecules diffuse from a region of high drug concentration to a region of low drug concentration [90,91]. Our own research in encapsulating cell-free expression systems (involving a multitude of small molecules) suggested that passive leakage of small molecules across double emulsions can only be curtailed when the small molecule concentrations across the middle phase of the emulsion are maintained—chemo-isotonic solution [35]. In simple terms, the chemical composition of the small molecules needs to be maintained across the middle phase. However, this strategy is counterproductive in the case of emulsions designed for polypharmaceutical applications. An alternative to this strategy is to employ the right surfactant, which can stabilize the emulsion and prevent the leakage of the encapsulated actives. For example, usage of ionic or non-ionic surfactants with excellent packing can prevent the collapse of the emulsions as well as prevent the easy diffusion of encapsulates. A wide variety of surfactants are available, both natural and synthetic. However, non-ionic surfactants are highly preferred over ionic surfactants as the latter are susceptible to charge-induced destabilizations [92]. Non-ionic surfactants such as PGPR and Tween/Span 80 are commonly used emulsifiers. In fact, PGPR is currently the most widely used surfactant to stabilize double emulsions with more than 70% of the cited works in this review employing this surfactant. Using this surfactant-stabilized complex emulsion, most of the researchers have opted to convert the complex emulsion into polymeric microcapsules. Indeed, the advantage of using complex emulsions is their ability to transform the middle oil layer into a polymeric membrane with interesting permeability properties [23,84,93]. Lee et al. (2021) reported a semipermeable polymeric poly(ethylene glycol) diacrylate membrane in W/O/W/O-type triple emulsion [93]. The researchers have successfully shown the semipermeable nature of the membrane—negatively charged small molecules are blocked while zwitterionic small molecules are permeable. Besides polymerizing the emulsions, it is also important to explore other strategies such as the usage of lipids as surfactants to prevent the leakage of small molecules. For example, lipid-based formulations are extensively used in drug delivery [94,95]. The use of lipids as surfactants in stabilizing double and complex emulsions has been continuously explored since the 20th century [96]. Rapid advancements in this field have allowed the stable formation of microfluidic-based liposomes with encapsulates involving small molecules like fluorescein (300 daltons), proteins like GFP (50 kDa), cellular machineries like cytoskeleton, small vesicles, and even cells (see Figure 3) [22,97,98,99,100,101,102]. Besides being biocompatible, lipids are easily available and can be altered to achieve targeted drug delivery [95,103,104]. In our experience, we have not observed leakage of these molecules unless the entire emulsion is completely destabilized.

Despite the advancements in surfactants and leakage studies, no prolonged shelf-life studies of complex emulsions have been carried out. Amongst the literature reviewed in this study, the stability of the complex emulsions is no more than a few hours [32,67], a few days [56,59], or a few weeks [81,82]. Longevity studies are necessary for commercial use cases of complex, emulsions-based drug delivery vehicles. Performing these studies and optimizing the leakage properties of the emulsions is a challenge that needs to be tackled.

(b)Enhanced Stability

Complex emulsions offer an advantageous platform for enhancing the stability of encapsulated drugs, protecting sensitive molecules from degradation by enzymes, pH, and other factors within the biological environment. However, the overall stability of the emulsions depends on the physical and chemical properties of materials used in the formulation as well as their interactions with each other [105]. An in-depth review on the factors affecting the stability of double emulsions is not within the scope of this review. However, the readers are directed to a review on double emulsions by Garti and Aserin [106] and a comprehensive review by Muschiolik and Dickinson on the stability and applications of double emulsions in the context of food systems [107]. For example, electrolytes, surfactants, and the interfacial tension across multiple different interfaces needs to be optimized to achieve longer-term stability. Recent studies have shown that electrolytes play a crucial role in emulsion stabilization. Kwak et al. (2022) have shown that electrolytes such as NaCl, KCl, and MgCl_2_ positively impact the stability of the double emulsions whose inner droplet fraction is 50% of the double emulsion. In fact, it is reported that even the release properties of the double emulsions can be altered with the presence of electrolytes such as NaCl [108]. The authors argued that the gelling behaviour of gelatine coupled with NaCl interaction with PGPR could have reduced the Laplace pressure and increased surfactant packing—allowing for a stable emulsion—at least for the length of the study, 1 month. Lee et al. (2023) presented NaCl and low inner droplet fraction as a successful strategy to stabilize double emulsions for long-term applications [109]. However, osmotic pressure induced by these electrolytes also plays a major role in stabilizing the emulsions. It is well known that water can diffuse across the oil layer in double and complex emulsions, maintaining isosmotic conditions across the middle oil phase [91]. The theoretical and experimental investigation of the structural evolution of W/O/W double emulsions triggered by the diffusion of hydrophilic species is examined by Sameh et al. (2012) [110]. The study delved into the diffusion process, which encompassed the concurrent outward permeation of the initially encapsulated species and the inward diffusion of the osmotic regulator. The findings presented a substantial body of evidence suggesting that the evolution pattern (either swelling or deswelling) primarily relies on the permeation ratio. They proposed that the observations held true regardless of the specific chemical characteristics of the species involved in the transfer. Regarding the role of electrolytes in reducing the charge-based repulsion, Zhang et al. (2014) have shown that NaCl can reduce the charge repulsions between the polar heads of the surfactants and help in better packing the surfactant for emulsion longevity, up to 2 months [111]. Thus, a combination of electrolytes and surfactants could be the answer to avoid drug leakage.

Surfactants are one other important factor in stabilizing emulsions. While PGPR is the most widely used surfactant utilized in creating complex emulsions and facilitating their subsequent conversion into microcapsules, other categories of surfactants such as pegylated block co-polymers, lipids, and nanoparticles are explored for stabilizing complex emulsions [112]. For example, we and others have implemented phospholipids as double [22] and complex emulsion [71] stabilizers. Furthermore, nanoparticle-based stabilization is also considered—Pickering emulsions are emulsions stabilized via nano/microparticle self-assembly along the fluid interface. We used graphitic carbon nitride nanoparticles to stabilize Pickering emulsions of single, double, and complex emulsions in nature, further converting them into polymeric microcapsules [20,78]. Using nanoparticles can have further advantages beyond emulsion stability; they can be used as probes to locate the emulsions using MRI or ultrasound. Even more, they can be used to deliver and trigger drug release [85].

Despite these advancements, extending the complex emulsion shelf-life beyond 6 months to one year has not been proven and, thus, needs to be evaluated in future research.

(c)Formulation

To be used as a formulation, complex emulsions need to remain intact throughout their shelf life and be able to deliver the requisite drug amount in the desired dosage upon application.

Rheology of the emulsions plays a crucial role in the stability of the emulsions as well as the dosage. Sedimentation or flocculation will lead to inconsistent dosage; thus, it is important to alter the formulation of the dosage form to avoid such irregularities. One way to achieve an accurate dosage is to achieve homogeneous suspension of emulsions throughout the solution. Aggregation/flocculation could lead to coalescence and breaking of the emulsion, creating large, distinct, separate phases. Thus, rheology of the formulation plays a crucial role. While the use of certain surfactants and charge-based repulsion strategies can prevent coalescence, flocculation or creaming is difficult to prevent, especially in double or complex emulsions where the middle oil phase is less dense resulting in creaming (emulsions exists as a layer above the continuous phase). To avoid this, for example, in the case of double emulsion, either the density of the inner aqueous solution or the outer continuous aqueous solution should be made viscose. This allows for a lower creaming effect and accurate dosage per volume. Flaiz et al. (2016) have shown that gelled double emulsions have higher retention and remain stable for longer periods of time over their non-gelled counter parts [51,108]. They have shown that both salt and gelatine up to 10% can provide stability and content retention of 100% for the tested 22 days. However, having a dense inner aqueous solution leads to a sinking effect where the inner droplet settles at the bottom of the middle oil droplet (see Figure 6 for the schematic representation—rheology, under formulation). This behaviour is not unusual as water itself is denser than most oils used in double emulsion preparation (ρ_W1_ > ρ_O_). However, this phenomenon leads to partial dewetting and membrane formation. Even more, its can act as a nucleation point for complete dewetting [25]. Care must be taken in the choice of surfactants to prevent leakage and destabilization of emulsions due to dewetting because of the sinking effect [78]. This presents a clear solution for implementing a high-viscosity outer continuous phase (η_W2_ > η_W1_) to achieve uniform emulsion dispersion and accurate dosage in contrast to denser, viscous inner solution, which would result in added pressure on the thin surfactant membrane.

In addition to rheology, a significant challenge lies in identifying the suitable combinations of drug molecules that can be utilized together within a complex emulsion. Drug incompatibilities and drug–drug interactions represent well-documented hurdles in the development of pharmaceutical dosage forms and in the context of polypharmacy [2,114]. Drug incompatibilities pertain to the physical and chemical reactions of drugs during the formulation stage and prior to patient administration, while drug–drug interactions typically refer to interactions between drugs in vivo [114]. Complex emulsions are heralded as potential solutions for polypharmacy. Strategies such as segregating drugs into distinct compartments within the emulsion to prevent mixing and orchestrating the sequential release of drugs in vivo must be the key considerations during the formulation process. As mentioned earlier, careful choice of surfactants can prevent leakage between the compartments (see Figure 6 under formulations). Leister et al. (2023) have presented coalescence pathways within and amongst complex emulsions to determine the dominant pathways in the presence of varying sets of surfactants. This study presented a comprehensive methodology to evaluate the coalescence pathways for a stable, complex, emulsion-based formulation. Initially, the authors specified a combination of coalescence pathways and a specific set of surfactants to achieve excellent stability for a period of 24 h [115]. While sequential release is a challenge, it is not impossible to achieve. As shown in Figure 5c, it is possible to achieve sequential drug release by carefully choosing the materials used to form the compartments. In the results, the authors have employed PLA as a hydrolysable polymer and PCL as a non-hydrolysable shell [67]. Similarly, it is also possible to alter both the shell properties and the inner compartments. For example, through selective formation of hydrogels or liquid–liquid phase separation within one of the inner compartments, one could lower the rate of drug release, thus reducing drug interactions despite simultaneous release of contents from all the compartments.

(d)Scalability

One of the challenges historically associated with microfluidic technologies has been scaling up production to meet commercial needs. Recent research in this direction has lead to the development of off-the-shelf device designs that can produce complex emulsions in the desired pattern and in relatively higher volumes. Guo et al. (2023) has shown that complex emulsions can be 3D printed in desired patterns [74]. Furthermore, the authors have shown the device’s versatility in achieving spheroids within complex emulsions [116]. It is essential to note that manufacturing double emulsions or complex emulsions containing identical compartments at high throughput differs from producing those with differing compartments. Creating complex emulsions with compartments housing diverse actives or components necessitates the incorporation of specialized channels for each encapsulate within the microfluidic device. This requirement entails the development of a novel device(s), featuring a single inlet, dual inlets, or triple inlets, to enable the fabrication of intricate nested emulsions comprising one to three inner droplets with varying encapsulations. Thus, scalability in producing different types of complex emulsions involves developing designs for each one of them. This presents a significant challenge in the research and development of scalable microfluidic devices. Perhaps this is where parallelization of the microfluidic devices might find its place in large-scale production of microfluidic devices. Scalable production of single emulsions is achieved through stackable microfluidic design with a production rate up to 1 L/h [117,118,119]. In the case of higher order emulsions, one such device design was proposed by Nisisako et al., 2012 (see Figure 6, parallelization under scalability) [113]. The authors can manufacture single, Janus, double, and triple emulsions in large volumes as high as 180 mL/h. This volume is quite large considering the traditional microfluidic production rates are 0.1 to 10 mL/h, which is much less than commercial-scale requirements, L/h. A kilo-per-day, single-core double emulsions was achieved by the Wietz group using a 3D PDMS device [120]. Despite the steady increase in scale-up methodologies, the research is still in its infant stages with limited or no research in the high-throughput production of complex emulsions beyond double emulsions [11,121].

## 5. Future Directions and Conclusions

Polypharmacy, defined as the simultaneous use of multiple medications by a single patient, particularly among the elderly and individuals with multimorbidity, poses significant healthcare challenges [2]. Most importantly, the usage of multiple medication is necessary to treat underlying multimorbidity. Thus, polypharmacy is become a necessary evil [122]. In the context of polypharmacy, the risk of adverse drug reactions (ADRs) escalates significantly as the number of medications increases. This heightened risk is due to the complex pharmacokinetic and pharmacodynamic interactions that can occur when multiple drugs are simultaneously present in the body. Another important consequence of polypharmacy is drug–drug interactions. Drug–drug interactions occur when the effect of one drug is altered by the presence of another. These interactions can lead to reduced efficacy of medications or increase the risk of ADRs. In polypharmacy, the complexity of managing multiple medications significantly increases the likelihood of such interactions. Pharmacokinetic interactions may involve changes in the absorption, distribution, metabolism, or excretion of drugs, while pharmacodynamic interactions may result in additive, synergistic, or antagonistic effects. Identifying and managing drug–drug interactions are critical components of polypharmacy stewardship, requiring a detailed understanding of the drugs involved, their mechanisms of action, and their pharmacological profiles [62,123].

Apart from the usual side effects of medication, drug-interaction-induced side effects can range from mild discomfort to severe, life-threatening conditions, impacting patient health and quality of life. Such issues force the patient towards medication non-adherence; the failure to take medications as prescribed is a significant challenge in the management of polypharmacy [4,124]. The complexity of polypharmacy regimens, with multiple dosing schedules and potential side effects, can overwhelm patients, leading to intentional or unintentional non-adherence. Non-adherence can compromise the overall effectiveness of treatment, resulting in suboptimal health outcomes, increased healthcare utilization, and higher costs. Strategies to improve adherence in the context of polypharmacy include simplifying medication regimens through polypharmacy stewardship, using fixed-dose combination products when possible, employing reminder systems, and ensuring patient education and engagement in the decision-making process regarding their treatment plans. However, one could also include developing new dosage forms as an effective strategy. With persisting knowledge on drug–drug interactions (i.e., pharmacokinetics), new dosage forms can be developed to prevent drug–drug interactions, avoid multiple-pill prescriptions, simplify regimens, and, thus, improve the quality of life of the patients together with improved treatment adherence.

### 5.1. Role of Complex Emulsions in Simplifying Regimens

Complex emulsions could play a pivotal role in streamlining medication regimens by encapsulating multiple therapeutic agents within a single delivery vehicle. This innovative approach significantly simplifies the management of polypharmacy. By combining medications that would typically be taken separately into one formulation, complex emulsions reduce the number of doses a patient must remember to take, thereby decreasing the complexity of their medication schedules. This simplification can lead to improved adherence, reduced likelihood of dosing errors, and overall better patient outcomes. One of the inherent advantages of complex emulsions is their capacity to minimize drug–drug interactions, a common concern in polypharmacy. By physically segregating different drugs within separate compartments of the emulsion, the direct interaction between drugs is prevented, thus reducing the potential for adverse pharmacokinetic and pharmacodynamic interactions. This compartmentalization ensures that each drug can exert its therapeutic effect without interference from or to other drugs present in the system. Additionally, tailored drug release capabilities of complex emulsions can further diminish the risk of interactions by carefully timing the release of each drug to avoid concurrent peak plasma concentrations [67,86]. Complex emulsions offer the flexibility to design tailored drug release profiles, ensuring that each encapsulated drug is released at the optimal rate and timing for its intended therapeutic effect. Through the manipulation of emulsion composition, including the choice of oil phase, surfactants, and polymers, it is possible to create formulations that provide sustained, delayed, or triggered drug release. This level of control allows for the synchronization of drug release with the body’s circadian rhythms or disease-specific demands, enhancing the efficacy of the treatment while minimizing side effects.

### 5.2. Major Hurdles

Stability, passive diffusion of drugs, pharmaceutical analysis of the dosage form, and, finally, large-scale production and the challenges that need to be tackled to realize complex emulsions as a pharmaceutical formulation. This review clearly presented all the four challenges in detail under Section 4. Complex emulsions with 50% or lesser inner droplet volume to overall emulsion volume are more stable; using NaCl and MgCl_2_ as electrolytes can also ensure excellent surfactant packing, thus avoiding leakage of the actives [108,109]. Furthermore, maintaining the osmotic pressure across the emulsion is highly desirable to prevent coalescence and associated drug–drug interactions and emulsion destabilization [37,91]. Maintaining these conditions and performing prolonged stability studies beyond six months to one year is highly recommended for long-lasting shelf-life of the complex, emulsion-based products. Furthermore, developing instrumentation for large-scale production of these emulsions will be challenging with most progress limited to double emulsions alone [121,125]. Thus, more research in the direction of developing new instrumentation and devices towards higher order complex emulsions needs to be implemented. Introducing a new dosage form requires a comprehensive testing program to evaluate dosage-form efficacy. These tests should encompass a wide variety of suitability studies. For example, in any given formulation, the complex emulsions are typically a micrometer in size, thus limited to applications other than intravenous administration. This allows for evaluating these complex, emulsion-based dosage forms, similar to traditional topical and oral dosage forms, i.e., creams and pills. This includes designing in vitro digestion and drug release studies using dissolution apparatus mimicking physiological conditions [126,127], in vivo pharmacokinetic studies to understand the plasma concentrations of each drug, toxicology studies to evaluate any adverse drug–drug interactions, pharmacodynamic studies, patient compliance and adherence studies, and cost-effectiveness analysis. By undertaking this comprehensive testing program, researchers and manufacturers can optimize complex emulsions for polypharmacy management, ultimately improving patient outcomes and enhancing the usefulness of these innovative formulations. Addressing the polypharmacy problem entails a multifaceted approach that encompasses the identification and mitigation of ADRs and drug–drug interactions, along with interventions designed to enhance medication adherence. The integration of complex emulsions into drug delivery strategies represents a significant advancement in addressing the challenges of polypharmacy. By offering simplified regimens, reducing the risk of drug interactions, and enabling the customization of drug release profiles, complex emulsions hold the promise of enhancing patient care and optimizing therapeutic outcomes.

Looking to the future, the continuous advancement of microfluidic technologies and the expansion of polypharmacy stewardship play pivotal roles in navigating the evolving landscape of modern pharmacotherapy. The integration of these innovations with digital health technologies is essential for enabling real-time monitoring, data-driven analyses, and personalized healthcare interventions. While future challenges like regulatory intricacies, the need for additional clinical validation, and incorporating new technologies into existing healthcare frameworks may arise, these obstacles also present opportunities for innovation, collaboration, and reimagining drug delivery and medication management practices. Embracing these advancements and fostering a culture of continual improvement will be transformative in shaping the future of healthcare delivery and enhancing patient outcomes.

## Figures and Tables

**Figure 1 pharmaceutics-16-00707-f001:**
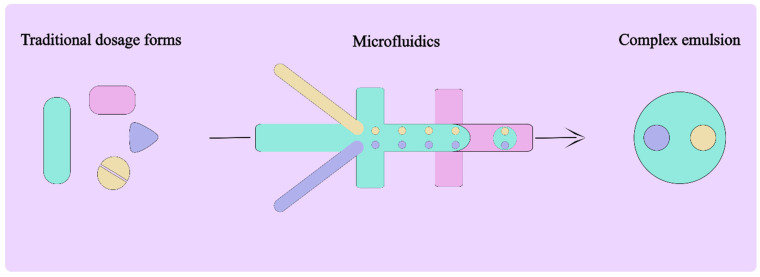
Transformation of traditional, multiple-dosage-form prescription into a single dosage form using microfluidics. In the schematic above, each active ingredient/dosage form is depicted using different colours. These active ingredient solutions can be transitioned into a complex emulsion with each ingredient occupying different compartments that form the complex emulsion. In this case, four different active ingredients can be encapsulated in a single complex emulsion.

**Figure 2 pharmaceutics-16-00707-f002:**
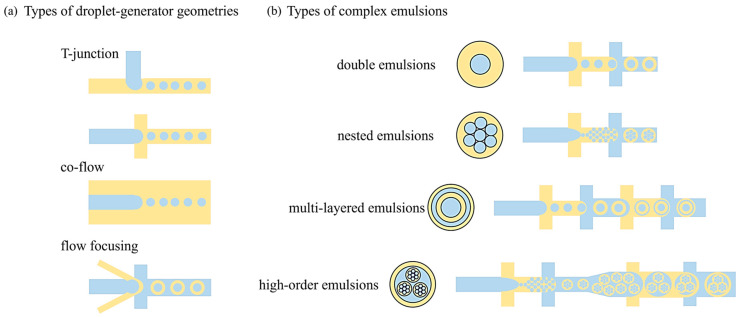
Microfluidic flow geometries and emulsion types. (**a**) Schematic representation of different microfluidic geometries typically used to produce droplets—T-junction/co-flow/flow-focusing methodologies. (**b**) Different types of complex emulsions and schematic representations of their production methods using PDMS-based microfluidics. The blue colour and yellow colours represent different immiscible fluids, i.e., aqueous or oil or gas.

**Figure 3 pharmaceutics-16-00707-f003:**
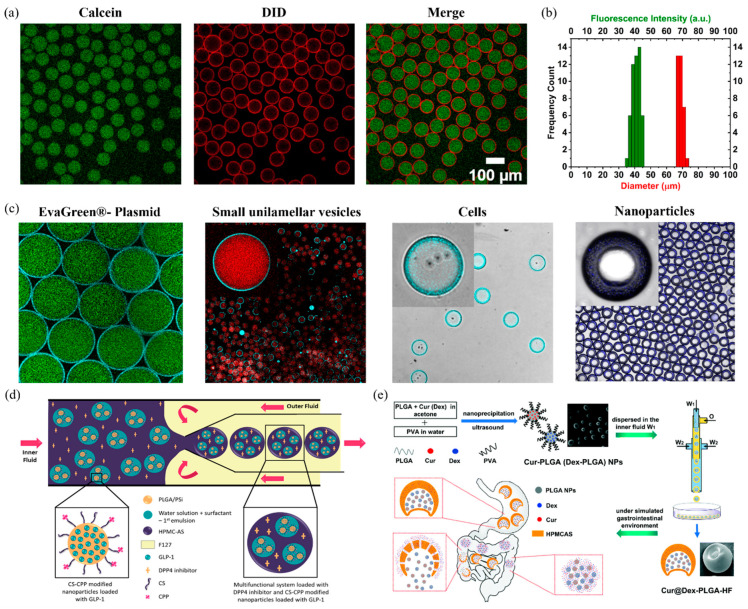
Double-emulsion-type complex emulsion production using microfluidics. (**a**,**b**) Double-emulsion-templated production of liposomes with high precision in size uniformity and encapsulation efficiency (Reprinted/adapted with permission from Ref. [22]. 2021, Springer Nature)”. (**c**) Highly uniform liposomes containing different encapsulated; from left to right: plasmid DNA, small unilamellar vesicles, cells, (Reprinted/adapted with permission from Ref. [22]. 2021, Springer Nature) and double emulsions with graphitic carbon nitride nanoparticles within the hydrophobic phase produced using PDMS-based microfluidics (Reprinted/adapted with permission from Ref. [20]. 2020, John Wiley and Sons). (**d**) Formation of microfluidic W/O/W using prefabricated W/O emulsion (mechanical agitation) with dual drug encapsulation for oral drug delivery (Reprinted/adapted with permission from Ref. [24]. 2015, American Chemical Society). (**e**) Microfluidic production of W/O/W emulsions with dual drug encapsulation along with drug-loaded nanoparticles for site-specific release properties after polymerization to treat inflammatory bowel disease delivery (Reprinted/adapted with permission from Ref. [32]. 2021, Royal Society of Chemistry).

**Figure 5 pharmaceutics-16-00707-f005:**
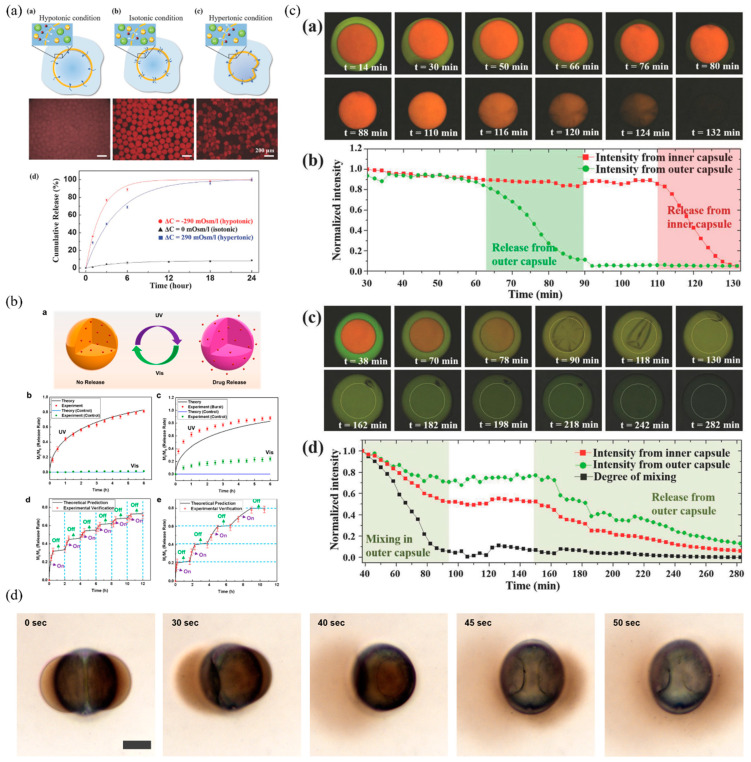
Drug release kinetics in double and complex emulsions. (**a**) Osmotic-pressure-induced drug release from W/O/W double-emulsion-based PLGA drug delivery vehicles (Reprinted/adapted with permission from Ref. [81]. 2017, Wiley and Sons). (**b**) Photo-switchable drug delivery carriers for programmable drug delivery in the presence of UV illumination (Reprinted/adapted with permission from Ref. [86]. 2023, American Chemical Society). (**c**) Layered-emulsion-based drug delivery vehicles showing both sequential and simultaneous dual drug delivery (Reprinted/adapted with permission from Ref. [67]. 2018, Wiley and Sons). (**d**) Osmotic-pressure-induced burst release of gold nanoparticles from polymerized nested emulsions (Reprinted/adapted with permission from Ref. [25]. 2022, Wiley and Sons).

**Figure 6 pharmaceutics-16-00707-f006:**
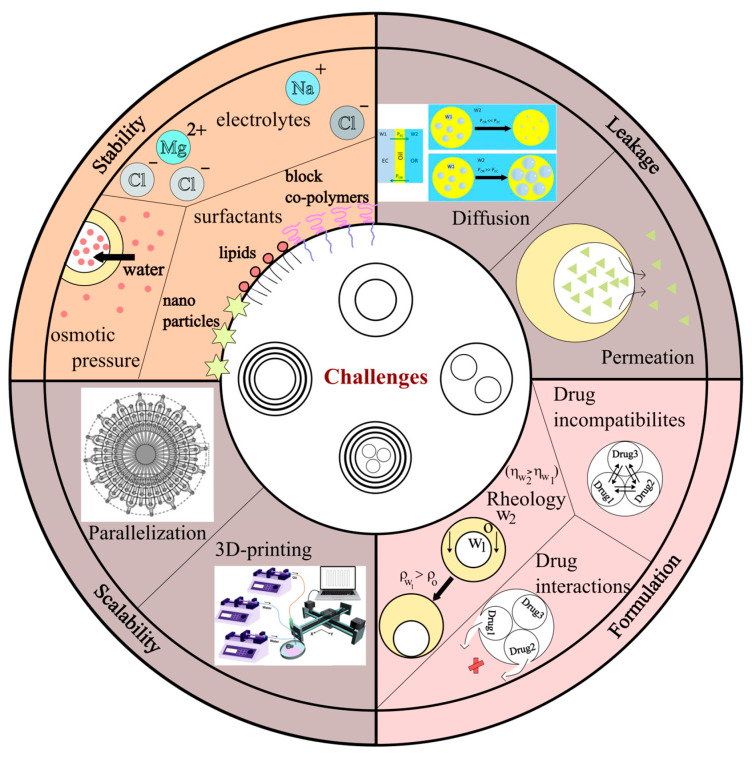
Challenges in addressing complex emulsions as a pharmaceutical formulation. Stability of the emulsions dictated by the concentrations of electrolytes and surfactants inducing emulsion destabilization via coalescence and osmotic pressure differences. Leakage of drugs from the emulsions renders the applications of the complex emulsions, especially passive leakage via diffusion and permeation to surfactant membranes. Emulsion formulation including the rheology for enhanced stability and choice of drug components to prevent adverse reactions is essential. Scalability: industrial-scale production is necessary for wider utilization and applications of the complex emulsions, which could be achieved through parallelization ((Reprinted/adapted with permission from Ref. [113]. 2012, Royal Society of Chemistry) of the devices or 3D printing methodologies ((Reprinted/adapted with permission from Ref. [65]. 2018, Elsevier).

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
