# Peer review of "Complex Emulsions as an Innovative Pharmaceutical Dosage form in Addressing the Issues of Multi-Drug Therapy and Polypharmacy Challenges"

_pharmaceutics, 2024, doi:10.3390/pharmaceutics16060707_

Round 1

Reviewer 1 Report

Comments and Suggestions for Authors

The work by Naresh Yandrapalli concerns complex emulsions as an innovative pharmaceutical dosage form in addressing the issues of multi-drug therapy and polypharmacy challenges. The topic of this manuscript is important and current, and results could be interesting for readers. However, some changes have to be entered into the revised version of the manuscript before it can be further processed:

1.     Chapters 2 and 3 lack information about what the application of a given method brings, in particular for the specific application examples cited.

2.     Tables 1 and 2 lack information about the advantages of using a given formulation and emulsion bases

3.     Chapter 4 lacks numerical data to confirm the statements, e.g. how much does the stability of the formulation increase and under what conditions. The same for the remaining examples

4.     Chapter 5 should be supplemented with a presentation of what tests should be completed to increase the usefulness of the described formulations

Author Response

The author thanks the reviewer 1 for their constructive comments. Responses are provided in a point-by-point manner in the attached file

Reviewer 2 Report

Comments and Suggestions for Authors

This paper reviews original aspects related to double emulsions, intimately related to their formulation processes.

A special care is brought on non-conventional processes and structures, proposes a description of the relationship between complex structure, process and drug release profiles.

As multiple emulsions are particular systems made with several phases and structures, the co-encapsulation of hydrophilic and/or lipophilic molecules is an important point, very specific to this system. This point is also reviewed, with an updated and suitable literature as reference.

Finally, the last and expected chapter presents the release mechanisms, and potential stimuli-responsive effect.

To conclude, this review appears very comprehensive and original and I recommend its publication as it is

Author Response

I am glad to hear the positive responses from the Reviewer 2. I am very thankful and encouraged from their support.

Round 2

Reviewer 1 Report

Comments and Suggestions for Authors

Please incorporate the revised Table 1 from the review response into the revised manuscript

Author Response

The Table 1 is included in the revised Manuscript.

Thank you for the support and helpful suggestions.
